# Innate-Values-driven Reinforcement Learning

## Abstract

Innate values describe agents' intrinsic motivations, which reflect their inherent interests and preferences for pursuing goals and drive them to develop diverse skills that satisfy their various needs. Traditional reinforcement learning (RL) is learning from interaction based on the environment's feedback rewards. However, in real scenarios, the rewards are generated by agents' innate value systems, which differ vastly from individuals based on their needs and requirements. In other words, considering the AI agent as a self-organizing system, developing its awareness through balancing internal and external utilities based on its needs in different tasks is a crucial problem for individuals learning to support others and integrate community with safety and harmony in the long term. To address this gap, we propose a new RL model termed innate-values-driven RL (IVRL) based on combined motivations' models and expected utility theory to mimic its complex behaviors in the evolution through decision-making and learning. Then, we introduce two IVRL-based models: IV-DQN and IV-A2C. By comparing them with benchmark algorithms such as DQN, DDQN, A2C, and PPO in the Role-Playing Game (RPG) reinforcement learning test platform VIZDoom, we demonstrated that the IVRL-based models can help the agent rationally organize various needs, achieve better performance effectively.

## 1 Introduction

In natural systems, motivation is concerned explicitly with the activities of creatures that reflect the pursuit of a particular goal and form a meaningful unit of behavior in this function Heckhausen & Heckhausen (2018). From the neuroscience perspective, intrinsic motivation refers to an agent's spontaneous tendencies to be curious and interested, to seek out challenges, and to exercise and develop their skills and knowledge, even without operationally separable rewards Di Domenico & Ryan (2017). Furthermore, they describe incentives relating to an activity itself, and these incentives residing in pursuing an activity are intrinsic Barto (2013). Moreover, intrinsic motivations deriving from an activity may be driven primarily by interest or activity-specific incentives, depending on whether the object of an activity or its performance provides the main incentive Schiefele (1996). They also fall in the category of cognitive motivation theories, which include theories of the mind that tend to be abstracted from the biological system of the behaving organism Merrick (2013).

However, natural agents, like humans, often make decisions based on a blend of biological, social, and cognitive motivations, as elucidated by combined motivations' model like Maslow's Hierarchy of Needs Maslow (1958) and Alderfer's Existence-Relatedness-Growth (ERG) theory Alderfer (1972). Fig. 1 illustrates the five human agents with various personalities presenting different amounts of innate values and preferences based on five levels of Maslow's Hierarchy of Needs and Alderfer's ERG theory. On the other hand, the AI agent can be regarded as a self-organizing system that also presents various needs and motivations in its evolution through decision-making and learning to adapt to different scenarios and satisfy their needs Merrick & Maher (2009).

Many researchers regard motivated behavior as behavior that involves the assessment of the consequences of behavior through learned expectations, which makes motivation theories tend to be intimately linked to theories of learning and decision-making Baldassarre & Mirolli (2013). In particular, intrinsic motivation leads organisms to engage in exploration, play, strategies, and skills driven by expected rewards. The computational theory of reinforcement learning (RL) addresses how

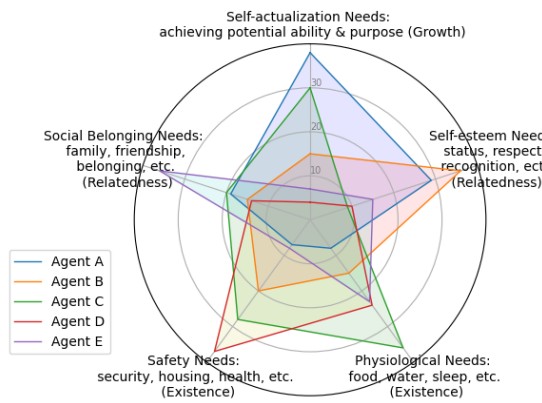

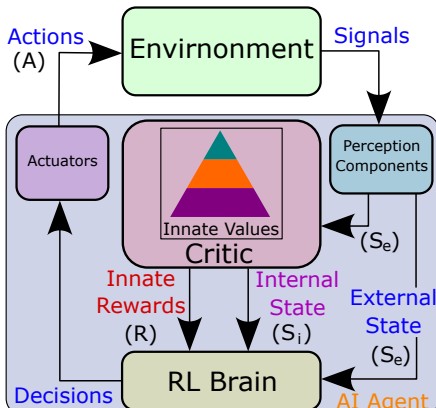

Figure 1: The illustration five human agents with various personalities presenting different amounts of innate values and preferences based on five levels of Maslow's Hierarchy of Needs and Alderfer's *Existence-Relatedness-Growth* (ERG) theory.

Figure 2: The illustration of the proposed innate-values-driven model.

predictive values can be learned and used to direct behavior, making RL naturally relevant to studying motivation. For example, development RL is concerned with using deep RL algorithms to tackle a developmental problem – the intrinsically motivated acquisition of open-ended repertoires of skills Colas et al. (2022).

In artificial intelligence, researchers propose various abstract computational structures to form the fundamental units of cognition and motivations, such as states, goals, actions, and strategies. For intrinsic motivation modeling, the approaches can be generally classified into three categories: prediction-based Schmidhuber (1991; 2010), novelty-based Marsland et al. (2000); Merrick & Maher (2009), and competence-based Barto et al. (2004); Schembri et al. (2007). Furthermore, the concept of intrinsic motivation was introduced in machine learning and robotics to develop artificial systems learning diverse skills autonomously Yang & Parasuraman (2020a; 2023; 2024). The idea is that intelligent machines and robots could autonomously acquire skills and knowledge under the guidance of intrinsic motivations and later exploit such knowledge and skills to accomplish tasks more efficiently and faster than if they had to acquire them from scratch Baldassarre & Mirolli (2013).

In other words, by investigating intrinsically motivated learning systems, we would clearly improve the utility and autonomy of intelligent artificial systems in dynamic, complex, and dangerous environments Yang & Parasuraman (2020b; 2021). Specifically, compared with the traditional RL model, intrinsically motivated RL refines it by dividing the environment into an external environment and an internal environment Aubret et al. (2019), which clearly generates all reward signals within the organism[1] Baldassarre & Mirolli (2013). Although the extrinsic reward signals are triggered by the objects and events of the external environment, and activities of the internal environment cause the intrinsic reward signals, it is hard to determine the complexity and variability of the intrinsic rewards (innate values) generating mechanism. Specifically, traditional RL model is learning from interaction based on the environment's feedback rewards. However, in real world, the rewards are generated by agents' innate value systems, which differ vastly from individuals based on their needs and requirements. Moreover, the AI agent can be regarded as a self-organizing system that also presents various needs and motivations in its evolution through decision-making and learning to adapt to different scenarios and satisfy those needs. The traditional RL can not reasonably explain its innate values and motivations nor provide a long-term model to support the AI agent's lifelong development.

To address those gaps, we introduce the innate-values-driven reinforcement learning (IVRL) model, which integrates combined motivations' models and expected utility theory to describe the complex behaviors in AI agents' adaptation and evolution. We formalize the innate values and derive the IVRL model, then propose two corresponding algorithms: IV-DQN and IV-A2C. Furthermore, we compare

---

[1]Here, the organism represents all the components of the internal environment in the AI agent.

them with benchmark RL algorithms such as DQN Mnih et al. (2015), DDQN Wang et al. (2016), A2C Mnih et al. (2016), and PPO Schulman et al. (2017) in the Role-Playing Game (RPG) RL test platform VIZDoom Kempka et al. (2016); Wydmuch et al. (2019). The results demonstrate that the proposed IVRL model can achieve convergence and adapt efficiently to complex and challenging tasks.

## 2 APPROACH OVERVIEW

We assume that all the AI agents (like robots) interact in the same working scenario, and their external environment includes all the other group members and mission setting. In contrast, the internal environment consists of individual perception components including various sensors (such as Lidar and camera), the critic module involving intrinsic motivation analysis and innate values generation, the RL brain making the decision based on the feedback of rewards and description of the current state (including internal and external) from the critic module, and actuators relating to all the manipulators and operators executing the RL brain's decisions as action sequence and strategies. Fig. 2 illustrates the proposed innate-values-driven model.

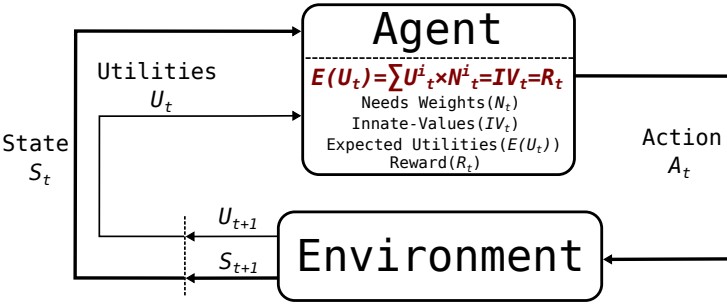

Figure 3: The illustration of the IVRL model based on Expected Utility Theory.

Compared with the traditional RL model, our model generates the input state and rewards from the critic module instead of directly from the environment, which means that the AI agent receives various utilities from the environment through executing an action or strategy in the IVRL model. Moreover, the individual needs to calculate innate values (expected utility) through its needs weights and current utilities and then select suitable actions or strategies to optimize or maximize its accumulated expected utility (Fig. 3). Specifically, we formalize the IVRL of an AI agent with an external environment using a Markov decision process (MDP) Puterman (2014). The MDP is defined by the tuple $\langle \mathcal{S}, \mathcal{A}, \mathcal{R}, \mathcal{T}, \gamma \rangle$ where $\mathcal{S}$ represents the finite sets of internal state $S_i{}^2$ and external states $S_e$. $\mathcal{A}$ represents a finite set of actions. The transition function $\mathcal{T}: \mathcal{S} \times \mathcal{A} \times \mathcal{S} \to [0, 1]$ determines the probability of a transition from any state $s \in \mathcal{S}$ to any state $s' \in \mathcal{S}$ given any possible action $a \in \mathcal{A}$. Assuming the critic function is $\mathcal{C}$, which describes the individual innate value model. The reward function $\mathcal{R} = \mathcal{C}(S_e) : \mathcal{S} \times \mathcal{A} \times \mathcal{S} \to \mathbb{R}$ defines the immediate and possibly stochastic innate reward $\mathcal{C}(S_e)$ that an agent would receive given that the agent executes action $a$ which in state $s$ and it is transitioned to state $s', \gamma \in [0, 1)$ the discount factor that balances the trade-off between innate immediate and future rewards.

### 2.1 THE EXPECTED UTILITY AND SOURCE OF RANDOMNESS

In the IVRL model, the reward is regarded as the *expected utility* Fishburn et al. (1979); Fishburn (1988) generated by the agent's utility values $u(x_k)$ and the corresponding probability $p_k$ (equation 1). It is equal to the sum of each category's needs weight $n_k$ times its current utility $u_k$ in the IVRL model (equation 3).

$$R_t = \sum_{i=1}^{k} u_k \times n_k = \mathbb{E}\left[U(p)\right] = \sum_{i=1}^{k} u(x_k)p_k \tag{1}$$

---

[2]The internal state $S_i$ describes an agent's innate value distribution and presents the dominant intrinsic motivation based on the external state $S_e$.

Furthermore, in the IVRL model, the randomness comes from three sources. The randomness in action is from the policy function: $A \sim \pi(\cdot|s)$; the needs weight function: $W \sim \omega(\cdot|s)$ makes the randomness of innate values; the state-transition function: $S' \sim p(\cdot|s, a)$ causes the randomness in state.

$$s_1 \longrightarrow a_1 \longrightarrow s_2 \longrightarrow a_2 \longrightarrow s_3 \longrightarrow a_3 \longrightarrow \cdots$$
$$w_1 \longrightarrow r_1 \qquad w_2 \longrightarrow r_2 \qquad w_3 \longrightarrow r_3$$

Figure 4: Illustration of the trajectory of state $S$, needs weight $W$, action $A$, and reward $R$ in the IVRL model.

Supposing at current state $s_t$ an agent has a needs weight matrix $N_t$ (equation 2) in a mission, which presents its innate value weights for different levels of needs. Correspondingly, it has a utility matrix $U_t$ (equation 2) for specific needs resulting from action $a_t$. Then, we can calculate its reward $R_t$ for $a_t$ through equation 3 at the state $s_t$.

$$N_t = \begin{bmatrix} n_{11} & n_{12} & \cdots & n_{1m} \\ n_{21} & n_{22} & \cdots & n_{2m} \\ \vdots & \vdots & \ddots & \vdots \\ n_{n1} & n_{n2} & \cdots & n_{nm} \end{bmatrix}; \quad U_t = \begin{bmatrix} u_{11} & u_{12} & \cdots & u_{1m} \\ u_{21} & u_{22} & \cdots & u_{2m} \\ \vdots & \vdots & \ddots & \vdots \\ u_{n1} & u_{n2} & \cdots & u_{nm} \end{bmatrix} \tag{2}$$

$$R_t = \sum_{i=1}^{m} \sum_{j=1}^{n} N_t \times U_t^T \tag{3}$$

In the process, the agent will first generate the needs weight and action based on the current state, then, according to the feedback utilities and the needs weights, calculate the current reward (expected utility) and iterate the process until the end of an episode. Fig. 4 illustrates the trajectory of state $S$, needs weight $W$, action $A$, and reward $R$, and Fig. 3 presents the corresponding IVRL model.

## 2.2 RANDOMNESS IN DISCOUNTED RETURNS

According to the above discussion, we define the discounted return $G$ at $t$ time as cumulative discounted rewards in the IVRL model (equation 4) and $\gamma$ is the discount factor.

$$G_t = R_t + \gamma R_{t+1} + \gamma^2 R_{t+2} + \cdots + \gamma^{n-t} R_n \tag{4}$$

At time $t$, the randomness of the return $G_t$ comes from the the rewards $R_t, \cdots, R_n$. Since the reward $R_t$ depends on the state $S_t$, action $A_t$, and needs weight $W_t$, the return $G_t$ also relies on them. Furthermore, we can describe their randomness as follows:

$$\text{State transition:} \quad \mathbb{P}[A = a|S = s, A = a] = p(s'|s, a); \tag{5}$$
$$\text{Needs weights function:} \quad \mathbb{P}[W = w|S = s] = \omega(w|s); \tag{6}$$
$$\text{Policy function:} \quad \mathbb{P}[A = a|S = s] = \pi(a|s). \tag{7}$$

## 2.3 ACTION-INNATE-VALUE FUNCTION

Based on the discounted return equation 4 and its random factors – equation 5, equation 6, and equation 7, we can define the *Action-Innate-Value* function as the expectation of the discounted return $G$ at $t$ time (equation 8).

$$Q_{\pi,\omega}(s_t, w_t, a_t) = \mathbb{E}[G_t|S_t = s_t, W_t = w_t, A_t = a_t] \tag{8}$$

$Q_{\pi,\omega}(s_t, w_t, a_t)$ describes the quality of the action $a_t$ taken by the agent in the state $s_t$, using the needs weight $w_t$ generating from the needs weight function $\omega$ as the innate value judgment to execute the policy $\pi$.

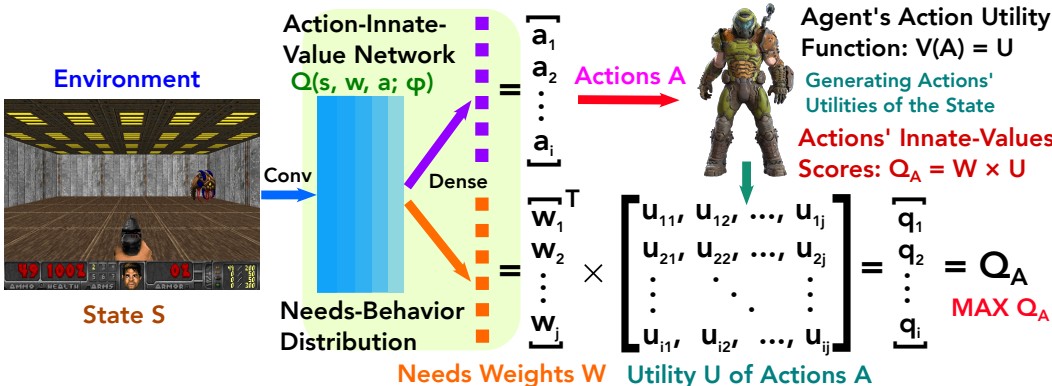

Figure 5: Illustration of the IV-DQN network generating Needs-Behavior distribution.

## 2.4 STATE-INNATE-VALUE FUNCTION

Furthermore, we can define the *State-Innate-Value* function as equation 9, which calculates the expectation of $Q_{\pi,\omega}(s_t, w_t, a_t)$ for action $A$ and reflects the situation in the state $s_t$ with the innate value judgment $w_t$.

$$V_\pi(s_t, w_t) = \mathbb{E}_A[Q_{\pi,\omega}(s_t, w_t, A)] \tag{9}$$

## 2.5 APPROXIMATE THE ACTION-INNATE-VALUE FUNCTION

The agent's goal is to interact with the environment by selecting actions to maximize future rewards based on its innate value judgment. We make the standard assumption that a factor of $\gamma$ per time-step discounts future rewards and define the future discounted return at time $t$ as equation 4. Moreover, we can define the optimal action-value function $Q^*(s, a, w)$ as the maximum expected return achievable by following any strategy after seeing some sequence $s$, making corresponding innate value judgment $w$, and then taking action $a$, where $\omega$ is a needs weight function describing sequences about innate value weights and $\pi$ is a policy mapping sequences to actions.

$$Q^*(s, w, a) = \max_{\omega, \pi} \mathbb{E}[G_t | S_t = s_t, W_t = w_t, A_t = a_t, \omega, \pi] \tag{10}$$

Since the optimal action-innate-value function obeys the Bellman equation, we can estimate the function by using the Bellman equation as an iterative update. This is based on the following intuition: if the optimal innate-value $Q^*(s', w', a')$ of sequence $s'$ at the next time-step was known for all possible actions $a'$ and needs weights $w'$, then the optimal strategy is to select the reasonable action $a'$ and rational innate value weight $w'$, maximising the expected innate value of $r + \gamma Q^*(s', w', a')$,

$$Q^*(s, w, a) = \mathbb{E}_{s' \sim \epsilon} \left[ r + \gamma \max_{w', a'} Q^*(s', w', a') \bigg| s, w, a \right] \tag{11}$$

Furthermore, the same as the DQN Mnih et al. (2015), we use a function approximator (equation 12) to estimate the action-innate-value function.

$$Q(s, w, a; \theta) \approx Q^*(s, w, a) \tag{12}$$

We refer to a neural network function approximator with weights $\theta$ as a Q-network. It can be trained by minimising a sequence of loss function $L_i(\theta_i)$ that changes at each iteration $i$,

$$L_i(\theta_i) = \mathbb{E}_{s,w,a \sim \sigma(\cdot)} \left[ (y_i - Q(s, w, a; \theta_i))^2 \right] \tag{13}$$

$$y_i = \mathbb{E}_{s' \sim \epsilon} \left[ r + \gamma \max_{w', a'} Q(s', w', a'; \theta_{i-1}) \bigg| s, w, a \right] \tag{14}$$

Where equation 14 is the target for iteration $i$ and $\sigma(s, w, a)$ is a probability distribution over sequences $s$, needs weights $w$, and action $a$ that we refer to as the *needs-behavior distribution*. We

---

**Algorithm 1:** Innate-Values-driven DQN (IV-DQN)

1   Initialize replay memory $\mathcal{D}$ to capacity N;

2   Initialize the action-innate-value function Q with random neural network weights;

3   **for** *each episode* **do**

4      **for** *each environment step t* **do**

5         With probability $\epsilon$ select a random action $a_t$ and $w_t$, otherwise select
         $a_t = \max_{w,a} Q(\phi(s), w, a; \theta)$;

6         Execute action $a_t$ in emulator, calculate reward $r_t = w_t \times u_t$ based on agent current needs weights
         $w_t$ and utilities $u_t$, and image $x_{t+1}$;

7         Set $s_{t+1} = s_t, a_t, w_t, x_{t+1}$ and preprocess $\phi_{t+1} = \phi(s_{t+1})$;

8         Store transition $(\phi_t, a_t, w_t, r_t, \phi_{t+1})$ in $\mathcal{D}$;

9         Sample random minibatch of transitions $(\phi_j, a_j, w_j, r_j, \phi_{j+1})$ from $\mathcal{D}$;

10         Set

$$y_j = \begin{cases} r_j, & \text{for terminal } \phi_{j+1}; \\ r_j + \gamma \max_{w,a} Q(\phi_{j+1}, w', a'; \theta), & \text{else.} \end{cases}$$

11         Perform a gradient descent step on $(y_j - Q(\phi_j, w_j, a_j; \theta))^2$ according to equation 15.

---

can approximate the gradient as follows:

$$\nabla_{\theta_i} L_i(\theta_i) = \mathbb{E}_{s,w,a\sim\sigma(\cdot); s'\sim\epsilon} \left[ \left( r + \gamma \max_{w',a'} Q(s', w', a'; \theta_{i-1}) - Q(s, w, a; \theta_i) \right) \nabla_{\theta_i} Q(s, w, a; \theta_i) \right] \tag{15}$$

Instead of computing the full expectations in the equation 15, stochastic gradient descent is usually computationally expedient to optimize the loss function. Here, the weights $\theta$ are updated after every time step, and single samples from the *needs-behavior distribution* $\sigma$ and the emulator $\epsilon$ replace the expectations, respectively.

Our approach is a model-free and off-policy algorithm, which learns about the greedy strategy $a = \max_{w,a} Q(s, w, a; \theta)$ following a *needs-behavior distribution* to ensure adequate state space exploration. Moreover, the *needs-behavior distribution* selects action based on an $\epsilon$-greedy strategy that follows the greedy strategy with probability $1 - \epsilon$ and selects a random action with probability $\epsilon$. Fig. 5 illustrates the action-innate-value network generating Needs-Behavior distribution.

Moreover, we utilize the *experience replay* technique Lin (1992), which stores the agent's experiences at each time-step, $e_t = (s_t, w_t, a_t, r_t, s_{t+1})$ in a data-set $\mathcal{D} = e_1, \ldots, e_N$, pooled over many episodes into a *replay memory*. During the algorithm's inner loop, we apply Q-learning updates, or minibatch updates, to samples of experience, $e \sim \mathcal{D}$, drawn at random from the pool of stored samples. After performing experience replay, the agent selects and executes an action according to an $\epsilon$-greedy policy, as we discussed. Since implementing arbitrary length histories as inputs to a neural network is difficult, we use a function $\phi$ to produce our action-innate-value Q-function. Alg. 1 presents the algorithm of the IV-DQN.

## 2.6   IVRL ADVANTAGE ACTOR-CRITIC (A2C) MODEL

Furthermore, we extend our IVRL method to the Advantage Actor-Critic (A2C) version. Specifically, our IV-A2C maintains a policy network $\pi(a_t|s_t; \theta)$, a needs network $\omega(w_t|s_t; \delta)$, and a utility value network $u(s_t, a_t; \varphi)$. Since the reward in each step is equal to the current utilities $u(s_t, a_t)$ multiplying the corresponding weight of needs, the state innated-values function can be approximated by presenting it as equation 16. Then, we can get the policy gradient equation 17 and needs gradient equation 18 of the equation 16 deriving $V(s; \theta, \delta)$ according to the *Multi-variable Chain Rule*, respectively. We can update the policy network $\theta$ and needs network $\delta$ by implementing policy gradient and needs gradient, and using the temporal difference (TD) to update the value network $\varphi$.

Figure 6: The architecture of the IVRL Actor-Critic model.

---

**Algorithm 2:** Innate-Values-driven Advantage Actor-Critic (IV-A2C)

1 Procedure *N-Step IVRL Advantage Actor-Critic*;
2 Start with policy model $\pi_\theta$, needs model $\omega_\delta$, and utility value model $V_\varphi$;
3 **for** *each episode* **do**
4     Generate an episode $s_0, a_0, w_0, u_0, \cdots, s_{T-1}, a_{T-1}, w_{T-1}, u_{T-1}$ following $\pi_\theta(\cdot)$ and $\omega_\delta(\cdot)$
5     **for** *each environment step t* **do**
6         $U_{end} = 0$;
7         if $t + N \geq T$ else $U_\varphi(s_{t+N})$;
8         $U_t = \gamma^N U_{end} + \sum_{k=0}^{N-1} \gamma^k (u_{t+k}$ if $(t + k < T)$ else $0)$;
9         $L(\theta) = \frac{1}{T} \sum_{t=0}^{T-1} (U_t - V_\varphi(s_t)) \cdot w_t \cdot \text{grad}(\pi_\theta)$;
10         $L(\delta) = \frac{1}{T} \sum_{t=0}^{T-1} (U_t - V_\varphi(s_t)) \cdot \text{grad}(w_t) \cdot \pi_\theta$;
11         $L(\varphi) = \frac{1}{T} \sum_{t=0}^{T-1} (U_t - V_\varphi(s_t))^2$;
12         Optimize $\pi_\theta$ using $\nabla L(\theta)$;
13         Optimize $\omega_\delta$ using $\nabla L(\delta)$;
14         Optimize $U_\varphi$ using $\nabla L(\varphi)$

---

$$
\begin{aligned}
V_{\pi,\omega}(s) &= \sum_{a_t, w_t} \pi(a_t|s_t) \cdot \omega(w_t|s_t) \cdot u(s_t, a_t) \\
&\approx \sum_{a, w} \pi(a_t|s_t; \theta) \cdot \omega(w_t|s_t; \delta) \cdot u(s_t, a_t; \varphi) = V(s; \theta, \delta, \varphi)
\end{aligned}
\tag{16}
$$

$$
\text{grad } V(a_t, \theta_t) = V_\theta(s; \theta, \delta, \varphi) = \frac{\partial V(s; \theta, \delta, \varphi)}{\partial \theta} = \frac{\partial \pi(a_t|s_t; \theta)}{\partial \theta} \cdot \omega(w_t|s_t; \delta) \cdot u(s_t, a_t; \varphi) \tag{17}
$$

$$
\text{grad } V(w_t, \delta_t) = V_\delta(s; \theta, \delta, \varphi) = \frac{\partial V(s; \theta, \delta, \varphi)}{\partial \delta} = \pi(a_t|s_t; \theta) \cdot \frac{\partial \omega(w_t|s_t; \delta)}{\partial \delta} \cdot u(s_t, a_t; \varphi) \tag{18}
$$

Using an estimate of the utility $u$ function as the baseline function, we subtract the V value term as the advantage value. Intuitively, this means how much better it is to take a specific action and a needs weight compared to the average, general action and the needs weights at the given state equation 19. Fig. 6 illustrates the architecture of the IVRL actor-critic version and Alg. 2 presents the algorithm of the IV-A2C.

$$
A(s_t, a_t) = U(s_t, a_t) - V(s_t) \tag{19}
$$

## 3 EXPERIMENTS

Since the IVRL model differs from the traditional RL model, it uses need weights and corresponding utilities from the internal and external environment to calculate innate value rewards. The traditional

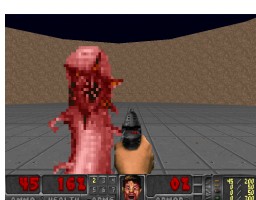 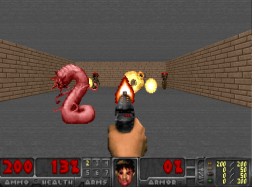 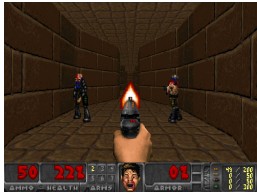 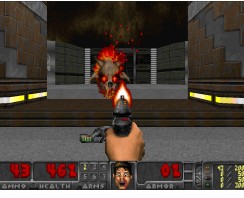

(a) Defend the Center     (b) Defend the Line     (c) Deadly Corridor     (d) Arena

Figure 7: The four scenarios of experiments in the VIZDoom

environment feedback-based reward RL platform can not be used in the IVRL experiments. Considering that the VIZDoom testbed Kempka et al. (2016); Wydmuch et al. (2019) can customize the experiment environment and define various utilities based on different tasks and cross-platform, we selected it to evaluate evaluate our IVRL model. We choose four scenarios: Defend the Center, Defend the Line, Deadly Corridor, and Arens (Fig. 7), and compare our models with several benchmark algorithms, such as DQN Mnih et al. (2015), DDQN Wang et al. (2016), A2C Mnih et al. (2016), and PPO Schulman et al. (2017). These models were trained on an NVIDIA GeForce RTX 3080Ti GPU with 16 GiB of RAM.

## 3.1 ENVIRONMENT SETTING

In our experiments, we define four categories of utilities (health points, amount of ammo, environment rewards, and number of killed enemies), presenting three different levels of needs: low-level safety and basic needs, medium-level recognition needs, and high-level achievement needs. When the agent executes an action, it receives all the corresponding innate utilities, such as health points and ammo costs, and external utilities, such as environment rewards (living time) and the number of killed enemies. At each step, the agent can calculate the rewards for the action by multiplying the current utilities and the needed weight for them. In our experiments, we assume that the agent has no bias or preference at the beginning of the game. Therefore, the initial needs weight for each utility category is 0.25, fixed in the benchmark DRL algorithms' training, such as DQN, DDQN, and PPO. For more details about the experiment code, please check the supplementary materials.

*a. Defend the Center – Fig. 7(a)*: For this scenario, the map is a large circle where the agent is in the middle, and monsters spawn along the wall. The agent's basic actions are turn-left, turn-right, and attack, and the action space is 8. It needs to survive in the scenario as long as possible.

*b. Defend the Line. – Fig. 7(b)*: The agent is located on a rectangular map, and the monsters are on the opposite side. Similar to the defend the center scenario, the agent needs to survive as long as possible. Its basic actions are move-left, move-right, turn-left, turn-right, and attack, and the action space is 32.

*c. Deadly Corridor. – Fig. 7(c)*: In this scenario, the map is a corridor. The agent is spawned at one end of the corridor, and a green vest is placed at the other end. Three pairs of monsters are placed on both sides of the corridor. The agent needs to pass the corridor and get the vest. Its basic actions are move-left, move-right, move-forward, turn-left, turn-right, and attack, and the action space is 64.

*d. Arena. – Fig. 7(d)*: This scenario is the most challenging map compared with the other three. The agent's start point is in the middle of the map, and it needs to eliminate various enemies to survive as long as possible. Its basic actions are move-left, move-right, move-forward, move-backward, turn-left, turn-right, and attack, and the action space is 128.

## 3.2 EVALUATION

The performance of the proposed IV-DQN and IV-A2C models is shown in the Fig. 8(a), 8(d), 8(g), and 8(j) demonstrate that IVRL models can achieve higher average scores than traditional RL benchmark methods. Especially for the IV-A2C algorithm, it presents more robust, stable, and efficient properties than other models.

Moreover, we analyze their corresponding tendencies in different scenarios to compare the needs weight differences between the IV-DQN and IV-A2C models. In the defend-the-center and defend-the-

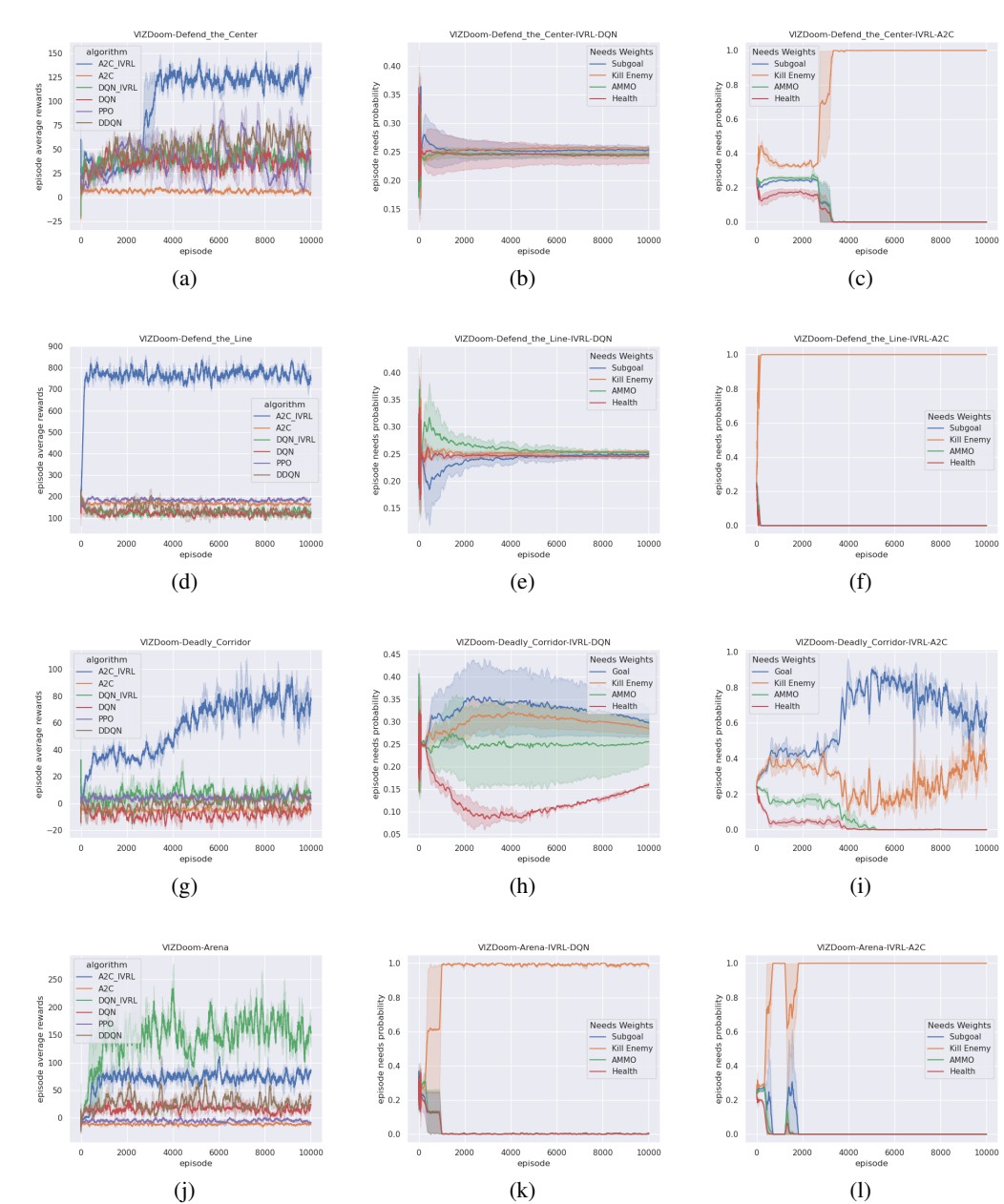

Figure 8: The performance comparison of IV-DQN and IV-A2C agents with DQN, DDQN, PPO, and A2C in the VIZDoom.

line experiments, each category of the need weight in the IV-DQN model does not split and converges to a specific range compared with its initial setting in our training (Fig. 8(b) and 8(e)). In contrast, the weights of health depletion, ammo cost, and sub-goal (environment rewards) shrink to approximately zero, and the weight of the number of killed enemies converges to one in the IV-A2C model. This means that the top priority of the IV-A2C agent is to eliminate all the threats or adversaries in those scenarios so that it can survive, which is similar to the Arena task. According to the performance in those three scenarios (Fig. 8(c), 8(f), and 8(l)), the IV-A2C agent represents the characteristics of bravery and fearlessness, much like the human hero in a real battle. However, in the deadly corridor mission, the needs weight of the task goal (getting the vest) becomes the main priority, and the killing enemy weight switches to the second for the IV-A2C agent (Fig. 8(i)). They converge to around 0.6

and 0.4, respectively. In training, by adjusting its different needs weights to maximize rewards, the IV-A2C agent develops various strategies and skills to kill the encounter adversaries and get the vast efficiently, much like a military spy.

In our experiments, we found that selecting the suitable utilities to consist of the agent innate-values system is critically important for building its reward mechanism, which decides the training speed and sample efficiency. Moreover, the difference in the selected utility might cause some irrelevant experiences to disrupt the learning process, and this perturbation leads to high oscillations of both innate-value rewards and needs weight. Furthermore, the IV-DQN performs better in the Arena than any other algorithm (Fig. 8(j)). However, in other experiments, the IV-A2C's performance is better than the IV-DQN. It reflects that, due to different task scenarios, the small perturbation introduced by the innate-values utilities may have made it difficult for the network weights in some topologies to reach convergence. Generally speaking, the performances of IV-DQN and IV-A2C are generally better than traditional A2C, DQN, and PPO.

The innate value system serves as a unique reward mechanism driving agents to develop diverse actions or strategies satisfying their various needs in the systems. It also builds different personalities and characteristics of agents in their interaction. From the environmental perspective, due to the various properties of the tasks, agents need to adjust their innate value system (needs weights) to adapt to different tasks' requirements. These experiences also shape their intrinsic values in the long term, similar to humans building value systems in their lives.

## 4 CONCLUSION

This paper introduces a new RL model from individual intrinsic motivations perspectives termed innate-values-driven reinforcement learning (IVRL). It is based on the expected utility theory to model mimicking the complex behaviors of agent interactions in its evolution. By adjusting needs weights in its innate-values system, it can adapt to different tasks representing corresponding characteristics to maximize the rewards efficiently. For theoretical derivation, we formulated the IVRL model and proposed two types of IVRL models: IV-DQN and IV-A2C. Furthermore, we compared them with benchmark algorithms such as DQN, DDQN, A2C, and PPO in the RPG reinforcement learning test platform VIZDoom. The results prove that rationally organizing various individual needs can effectively achieve better performance. Moreover, in the multi-agent setting, organizing agents with similar interests and innate values in the mission can optimize the group utilities and reduce costs effectively, just like "Birds of a feather flock together." in human society. Especially combined with AI agents' capacity to aid decision-making, it will open up new horizons in human-multi-agent collaboration. This potential is crucially essential in the context of interactions between human agents and intelligent agents when considering establishing stable and reliable relationships in their cooperation, particularly in adversarial and rescue mission environments.

For future work, we want to improve the IVRL further and develop a more comprehensive system to personalize individual characteristics to achieve various tasks testing in several standard MAS testbeds, such as StarCraft II, OpenAI Gym, Unity, etc. Especially in multi-object and multi-agent interaction scenarios, building the awareness of AI agents to balance the group utilities and system costs and satisfy group members' needs in their cooperation is a crucial problem for individuals learning to support their community and integrate human society in the long term. Moreover, integrating efficient deep RL algorithms with the IVRL can help agents evolve diverse skills to adapt to complex environments in MAS cooperation. Furthermore, implementing the IVRL in real-world systems, such as human-robot interaction, multi-robot systems, and self-driving cars, would be challenging and exciting.

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

# A APPENDIX

We provide the code of IV-QDN and IV-A2C models for the corresponding experiments. Please check the supplemental material.

