# OpenReview forum: "Innate-Values-driven Reinforcement Learning"
_ICLR.cc/2025/Conference — ICLR 2025 Conference Withdrawn Submission_

### Official Review · Reviewer_QeRg · 2024-10-16

**Soundness:** 3
**Presentation:** 3
**Contribution:** 3
**Rating:** 3
**Confidence:** 4

**Summary:**

The paper proposes a novel RL framework called **Innate-Values-Driven RL (IVRL)**, which combines innate values-based intrinsic motivations with traditional RL. It aims to enable agents to learn by balancing internal motivations and external rewards, mimicking human decision-making and behavior more closely. Based on the IVRL framework, two algorithms: IV-DQN, IV-A2C, are proposed. Experiments are conducted using the VizDoom benchmark, and the proposed algos outperformed standard RL algorithms (e.g., DQN, A2C, and PPO) and demonstrated better performance in complex, adaptive tasks.

**Strengths:**

- I pretty like the research that attempts to mimic human learning patterns to enhance AI systems. IVRL addresses the gap in RL by incorporating agents' intrinsic motivations, inspired by human decision-making models like Maslow's Hierarchy of Needs. This is a notable and interesting shift from traditional RL's focus purely on external rewards.

- The paper provides a clear and structured approach to calculating rewards by leveraging **expected utility theory**, allowing agents to maximize their overall satisfaction by considering both innate needs and external stimuli.

- The proposed framework can be extended beyond single-agent systems, indicating its potential for multi-agent environments, cooperative tasks, and even long-term learning scenarios.

**Weaknesses:**

Please refer to the questions part.

**Questions:**

Based on the presented paper, I've the following questions:

- There are no a well-organized related work section, as the authors claimed a new RL paradigm, a clear comparsion with exisiting work is essential.

- The method section seem a bit messy and can be orangnized better.

- As the paper indicates, the agent’s performance is highly sensitive to the selection of utility components and weight assignments, and small perturbations or poor choices could significantly affect learning efficiency and convergence. How can we solve this issue?

- I wonder if the authors used the `\citep{}` and `\citet{}` commands correctly, the citation format looks weird.

- In fact, intrinsic motivation (intrinsic rewards) has been widely studied in recently years, which can effectively solve sparse-rewards and hard-exploration tasks, such as ICM [1], RE3 [2], and E3B [3]. What‘s the key difference between intrinsically-motivated RL and IVRL?

- Regarding the question above, why the paper does not compare with these intrinsic reward methods?

- What's the key advantage of IVRL? or What can IVRL bring us? For example, solving extremely hard-exploration problems like intrinsic rewards. I recommend the authors to highlight this point. New paradigm is great, but the performance also matters.

- The experiments are only performed using the VizDoom, which is not sufficient. More envs like Procgen and MiniGrid should be evaluated.

- Detailed experimental settings should be added in the appendix.

- I recommend the authors to report experiment results using Rliable library [4].

- The IVRL framework introduces several additional layers of complexity, including the need-weight matrices and utility calculations, which may be difficult to implement or optimize in large-scale environments. Could u provide an efficiency analysis?

- I found a similar paper entitled "Innate-Values-driven Reinforcement Learning for Cooperative Multi-Agent Systems" [5], could u clarify the connections between it and this paper? [Paper Link]([Paper2_CMASDL_Workshop_AAAI24.pdf - Google 云端硬盘](https://drive.google.com/file/d/1gDoK_wgDdvAqEgm9aCF-66PKomypiDTo/view))


I'm happy to discuss these problems further with the authors.

# References

[1] Pathak D, Agrawal P, Efros A A, et al. Curiosity-driven exploration by self-supervised prediction[C]//International conference on machine learning. PMLR, 2017: 2778-2787.

[2] Seo Y, Chen L, Shin J, et al. State entropy maximization with random encoders for efficient exploration[C]//International Conference on Machine Learning. PMLR, 2021: 9443-9454.

[3] Henaff M, Raileanu R, Jiang M, et al. Exploration via elliptical episodic bonuses[J]. Advances in Neural Information Processing Systems, 2022, 35: 37631-37646.

[4] Agarwal R, Schwarzer M, Castro P S, et al. Deep reinforcement learning at the edge of the statistical precipice[J]. Advances in neural information processing systems, 2021, 34: 29304-29320.

[5] Yang Q. Innate-Values-driven Reinforcement Learning for Cooperative Multi-Agent Systems[J]. arXiv preprint arXiv:2401.05572, 2024.

---

### Official Review · Reviewer_w8z4 · 2024-10-30

**Soundness:** 1
**Presentation:** 1
**Contribution:** 2
**Rating:** 1
**Confidence:** 4

**Summary:**

The paper introduces a method that can augment existing RL algorithms such as DQN or A2C: innate-values-driven reinforcement learning (IVRL). Authors try to investigate intrinsic motivation based on "needs" (such as Maslow), which are represented by a utility function provided by a critic rather than the environment's rewards. Results show that using this augmentation technique can outperform non-innate-value augmented algorithms by a very large margin (approx. 2.5x to 5x).

**Strengths:**

- Interesting future work proposal

**Weaknesses:**

- Authors need to undertake serious restructuring of the paper's storyline. I feel like is it written with too complicated sentence, language and argumentation.
- The paper would benefit from drastic simplification and focus on the essentials. This is to a point where it affects the comprehensibility of the work and misunderstandings due to lack of information/clarification of key concepts that are not provided or introduced too late.
- Figures do not help to further understand/support what is being written
- References are often times too generic and do not contextualize the work
There is no related work section, which is crucial to understanding what the authors are trying to improve/continue/adapt.

**Questions:**

- The abstract is hard to read and comprehend:
1. Contrasting "environment's" with "real scenarios", both are environments when considering RL to solve them? This is confusing
2. The sentence starting with "In other words, considering ..." does not help clarify, specifically because it's too long and very hard to comprehend to a point where it's not understandable.
3. Reinforcement Learning (RL) has been abbreviated, but the term is still written in full towards the end.
-> I would suggest re-writing this in clear and simple language, with shorter sentences and a "golden thread".

- Introduction:
1. First sentence is not comprehendable.
2. In the second sentence, I don't think the "neuroscience" definition of intrinsic motivation is correct ("spontaneous tendencies to be curious and interested".
3. First paragraph is very hard to comprehend due to grammar, but also arguments formulated.
4. There is a mention of the "human agents" but it is unclear how Maslow's Hirarchy of Needs and Alderfer's ERG are represented in Figure 1.
5. Strong statement like "In particular, intrinsic motivation leads organisms to engage in exploration, play, strategies, and skills driven by expected rewards" need references.
6. This statement starts very generic with the term "artificial intelligence" but ends with features and specific aspects that are part of RL: In artificial intelligence, researchers propose various abstract computational structures to form the fundamental units of cognition and motivations, such as states, goals, actions, and strategies.
7. References for the "three categories of intrinsic motivation" are very generic.
8. How does "investigating" "clearly improve", just because I investigate something it is not improving anything other than my knowledge and understanding? In other words, by investigating intrinsically motivated learning systems, we would clearly improve the utility and autonomy of intelligent artificial systems in dynamic, complex, and dangerous environments Yang & Parasuraman (2020b; 2021).
9. many more statements and claims in this section do not make sense or hold incorrect information and or have wrong or too generic references.
-> I strongly suggest re-writing this section with a simple and clear story-line and sensful chains of arguments and references supporting and contextualizing the work that is being proposed in the rest of the paper.

- Approach Overview
1. The statement is "Compared with the traditional RL model, our model generates the input state and rewards from the critic module instead of directly from the environment,", however Figure 3 does not show that. Figure 3 illustrates the some utility function that is returned by the environment.
2. At this point I highly suggest to introduce the Utility function as this is crucial to understand the method.

- The Expected Utility and Source of Randomness
1. At this point we only know the innate reward C(Se), but we do not know what the "innate values" (line 164) are.
2. Line 174 - here we also do not know what the "needs weight" matrix is, this needs to be explained, otherwise this is not comprehendable.
-> Unfortunately, the structure, when and how information is introduced is quite diffuse, to a point where it is not possible to understand what the authors are trying to do.

- Experiments
1. We cannot assume that there is an "internal" environment.
2. Line 392 duplicate text: evaluate evaluate
3. While the environments have been customized for "IVRL" to work ("The traditional environment feedback-based reward RL platform can not be used in the IVRL experiments.") I suggest to add evaluation result from original authors and or others that have worked and on and evaluated VIZDoom
4. Results show that for episode average rewards:
On "Defend the Center", that IVRL achieves 2.5x performance VS baselines
On "Defend the Line", that IVRL achieves 4x performance VS baslines
On "Deadly Corridor", that IVRL achieves 4x performance VS baslines
On "Arena", that IVRL achieves 5x performance VS baselines
Please verify those findings and contrast against SOTA algorithms on the VIZDoom environment.
5. We do not know how the utility function modifies the reward structure and hence it is not clear if the comparison against the baseline is correct

- Conclusion
1. Claims in regards to Multi-Agent systems should probably be part of a discussions section, supported by reasoning of the results and findings on the single-agent experiments.

---

### Official Review · Reviewer_kUK5 · 2024-11-01

**Soundness:** 2
**Presentation:** 2
**Contribution:** 1
**Rating:** 3
**Confidence:** 4

**Summary:**

This paper introduces Innate Values-Driven Reinforcement Learning (IVRL), a framework that integrates intrinsic motivations through needs weights into traditional MDP-based RL algorithms such as DQN and A2C. Experimental evaluations in the VIZDoom environment demonstrate that IVRL-based models achieve performance improvements by effectively organizing and balancing various innate needs.

**Strengths:**

Interesting story telling:

The paper adopts a first-principles approach, drawing insightful parallels between RL agents and humans by emphasizing the role of intrinsic motivation. This foundation effectively supports the development of the Innate Values-Driven RL (IVRL) framework, providing a clear and engaging motivation.

Clarity:

The manuscript is well-organized and clearly written and the included case studies effectively illustrate the authors' key points. This clarity ensures that the work is easy to follow.

Robust Experimental Validation:

The experimental results demonstrate performance improvements of the IVRL-A2C model compared to benchmark algorithms such as DQN, DDQN, and PPO within the VIZDoom environment. Additionally, the innate value attributes of the agents in Doom scenarios are  logically sound, providing evidence of the model’s effectiveness.

**Weaknesses:**

Novelty：

The proposed approach appears to be an incremental advancement within the RL framework by incorporating innate values into the MDP. Specifically, the method introduces innate values through a needs weight mechanism and modifies existing RL algorithms such as DQN and A2C by integrating these needs weights. However, both DQN and A2C are well-established and widely-used algorithms in the RL community. The integration of needs weights, while useful, does not constitute a fundamentally novel approach, as it primarily extends existing methodologies rather than introducing a new paradigm or significantly advancing the state of the art.

Insufficient Explanation of Needs Weight:

The feasibility and implementation details of the needs weight function are not adequately explained. The authors briefly mention in Section 2.1 that the needs weight function W∼ω(⋅∣s) introduces randomness into innate values, and in the IVRL-A2C model (line 319), it is indicated that this is output by a network. However, given that needs weight is a crucial component of the model, the explanation provided is overly simplistic and lacks depth. A more comprehensive description of how the needs weight is generated, its underlying mechanisms, and how it influences the agent’s decision-making process is necessary to convincingly demonstrate its feasibility and effectiveness.

Experimental Needs polish:

The experimental section requires further development to provide robust validation of the proposed IVRL models. Currently, the authors compare IVRL-based models (IV-DQN and IV-A2C) only against baseline algorithms such as DQN, DDQN, and PPO on a single reinforcement learning test platform, VIZDoom. This limited scope does not sufficiently demonstrate the generalizability and effectiveness of the proposed methods. Additionally, sota algorithms like SAC[1] and A3C[2] are not included in the comparisons, which are important baselines in the RL field. Furthermore, evaluating the needs weight mechanism across multiple benchmarks and including more diverse and complex case studies, would provide stronger evidence of the model’s scalability and practical applicability. Without these additional experiments and comparisons, the claims regarding the superior performance and rational organization of needs remain unconvincing.

**Questions:**

Please check the weakness especially in the first and second point.

---

### Official Review · Reviewer_EswB · 2024-11-01

**Soundness:** 2
**Presentation:** 2
**Contribution:** 2
**Rating:** 3
**Confidence:** 5

**Summary:**

The paper introduces innate-values-driven RL, an RL framework designed to incorporate intrinsic motivations, reflecting agents' inherent needs and preferences. IVRL integrates motivation models and expected utility theory to simulate complex agent behaviors and balance internal and external utilities. Two IVRL-based models, IV-DQN and IV-A2C, are evaluated against benchmarks (DQN, DDQN, A2C, PPO) in the VIZDoom RPG platform, demonstrating improved performance in organizing diverse needs and achieving goal-oriented behavior.

**Strengths:**

The focus on intrinsic motivations addresses a critical area in RL, allowing agents to develop behaviors aligned with real-world-like needs and goals.

**Weaknesses:**

The role of the intrinsic reward is unclear, as it seems tasks could still be completed using only the total (external) reward. The intrinsic reward appears to break down tasks into hand-crafted levels rather than adding meaningful new motivation parts.

The primary contribution is limited, as the paper mainly applies DQN and A2C under the proposed reward structure without substantial novelty in methodology.

The text in Figure 8 is too small, making it difficult to interpret.

The paper lacks a dedicated related works section, which is essential for positioning the work within existing literature.

**Questions:**

I have no further questions.

---

### Note · Authors · 2024-11-12

**Comment:**

Thank you very much for those feedback. We will upgrade our paper according to constructive suggestions.

**Withdrawal Confirmation:**

I have read and agree with the venue's withdrawal policy on behalf of myself and my co-authors.